# Conditional CT Strategy—An Effective Tool to Reduce Negative Appendectomy Rate and the Overuse of the CT

**DOI:** 10.3390/jcm10112456

**Published:** 2021-06-01

**Authors:** Raminta Luksaite-Lukste, Ruta Kliokyte, Arturas Samuilis, Eugenijus Jasiunas, Martynas Luksta, Kestutis Strupas, Tomas Poskus

**Affiliations:** 1Clinic of Gastroenterology, Nephrourology and Surgery, Faculty of Medicine, Institute of Clinical Medicine, Vilnius University, LT-08661 Vilnius, Lithuania; martynas.luksta@santa.lt (M.L.); kestutis.strupas@santa.lt (K.S.); tomas.poskus@santa.lt (T.P.); 2Department of Radiology, Nuclear Medicine and Medical Physics, Faculty of Medicine, Institute of Biomedical Sciences, Vilnius University, LT-08661 Vilnius, Lithuania; ruta.kliokyte@santa.lt (R.K.); arturas.samuilis@santa.lt (A.S.); 3Centre of Informatics and Development, Vilnius University Hospital, Santara Clinics, LT-08661 Vilnius, Lithuania; eugenijus.jasiunas@santa.lt

**Keywords:** acute appendicitis, diagnostic imaging, negative appendectomy, CT scan, ultrasound

## Abstract

(1) Background: Diagnosis of acute appendicitis (AA) remains challenging; either computed tomography (CT) is universally used or negative appendectomy rates of up to 30% are reported. Transabdominal ultrasound (TUS) as the first-choice imaging modality might be useful in adult patients to reduce the need for CT scans while maintaining low negative appendectomy (NA) rates. The aim of this study was to report the results of the conditional CT strategy for the diagnosis of acute appendicitis. (2) Methods: All patients suspected of acute appendicitis were prospectively registered from 1 January 2016 to 31 December 2018. Data on their clinical, radiological and surgical outcomes are presented. (3) Results: A total of 1855 patients were enrolled in our study: 1206 (65.0%) were women, 649 (35.0%) were men, and the median age was 34 years (IQR, 24.5–51). TUS was performed in 1851 (99.8%) patients, and CT in 463 (25.0%) patients. Appendices were not visualized on TUS in 1320 patients (71.3%). Furthermore, 172 (37.1%) of 463 CTs were diagnosed with AA, 42 (9.1%) CTs revealed alternative emergency diagnosis and 249 (53.8%) CTs were normal. Overall, 519 (28.0%) patients were diagnosed with AA: 464 appendectomies and 27 diagnostic laparoscopies were performed. The NA rate was 4.2%. The sensitivity and specificity for TUS and CT are as follows: 71.4% and 96.2%; 93.8% and 93.6%. (4) Conclusion: A conditional CT strategy is effective in reducing NA rates and avoids unnecessary CT in a large proportion of patients. Observation and repeated TUS might be useful in unclear cases.

## 1. Introduction

Acute appendicitis (AA) is one of the most common causes of urgent surgery, and appendectomy, the gold standard for AA treatment, is the most frequently performed emergency surgery [1]. Even though it has been centuries since AA was described for the first time by James Parkinson in 1812, diagnostics remain challenging, and the negative appendectomy rate can still be as high as 30% due to similarities to other common causes of abdominal pain [2,3,4].

Clinical and laboratory scores—Alvarado score, Appendicitis Inflammatory Response score (AIR), Paediatric Appendicitis Score (PAS) and others—were designed to improve the diagnostic accuracy. However, none of these scores can be used to “rule in” the diagnosis of AA without further diagnostic testing and/or surgical assessment. For example, the Alvarado score is proven to be a useful diagnostic “rule out” score (cut-off point of <5), but when used as a sole decision criterion for surgery (cut-off point of 7) it produces negative appendectomy rates as high as 13.3–16.2%, suggesting it is not sufficiently accurate [5].

Transabdominal ultrasound (TUS) and computed tomography (CT) have been shown to reduce the number of delayed cases and negative appendectomies [2]. According to the World Society of Emergency Surgery (WSES) guidelines, the first-line radiology modality for suspected AA should be TUS [6]. While comparatively cheap, non-ionizing and accessible, it is also highly operator- and patient-dependent, and has shown mediocre sensitivity and specificity in AA diagnosis—86% and 81% accordingly [7]. In contrast, CT has higher sensitivity and specificity (around 90% and 94%), is less operator- and patient-dependent, and can better evaluate complications and detect possible alternative causes of abdominal pain. However, CT is more expensive and uses ionizing radiation, which should be avoided in pregnant women, children and young adult populations [4,7]. Our previous experience of clinical, laboratory and US-based diagnosis of AA has resulted in a negative appendectomy rate of 22.9% [8]. The aim of the present study was to analyze the effectiveness of the conditional CT-based strategy in AA diagnosis.

## 2. Materials and Methods

This observational cohort study was performed at Vilnius University Hospital’s Santara clinic tertiary university hospital from January 2016 to January 2019. All adult (>18 y) patients, admitted to the emergency department and presenting with symptoms suggestive of acute appendicitis after surgical consultation, were included in the study. The only exclusion criterion was pregnancy. After surgical consultation, all patients were referred for the TUS. If TUS was inconclusive and clinical suspicion of AA was still present, CT was performed. Patients were operated on only in cases when radiological evidence of AA was present. A flowchart of the conditional CT-based strategy is presented in Figure 1. Patient data entered into the prospectively maintained database included the following: age, sex, radiological diagnosis (ultrasound and/or computed tomography findings), treatment strategies, operative and histopathological findings. Additionally, CT images were re-evaluated by two board-certified radiologists. A final diagnosis was assigned to every patient by an expert panel, based on histopathology, imaging, surgical findings, clinical information, and at least 6 months of follow-up. The approval of Vilnius Region Bioethics Committee was acquired (approval number 2019/3-1107-610).

### 2.1. Radiological Methods

TUS was performed in the emergency department by board-certified radiologists, radiology residents and abdominal surgeons with varied experience in emergency settings. In most cases, ultrasound was done with a convex (2–5 MHz) probe followed by a more detailed examination of right iliac fossa with a high-frequency linear probe using the graded compression technique. Four different ultrasound machines were used during the study period: Toshiba Aplio 500 Tokyo, Japan, GE Logiq S8 Milwuakee, WI, US, GE Logiq 9 Milwuakee, WI, US and Toshiba Xario 400 Tokyo, Japan. 

If the appendix was visualized on TUS it was classified into the following groups:

Normal appendix: diameter of appendix <7 mm, wall thickness of the appendix <2 mm, compressible appendix without secondary findings of free fluid in right iliac fossa, lymphadenopathy, infiltration of surrounding tissue.

AA: diameter of appendix ≥7 mm, wall thickness of the appendix ≥2 mm, uncompressible appendix with secondary findings of free fluid in right iliac fossa, lymphadenopathy, infiltration of surrounding tissue.

Probable AA: diameter of appendix ~7 mm (or less), wall thickness of the appendix ~2 mm (or less), compressible/partially compressible appendix with or without secondary findings of free fluid in right iliac fossa, lymphadenopathy, infiltration of surrounding tissue.

CT scans were performed using GE Discovery 750 HD (128 slices). Scans were assessed by board-certified radiologists specializing in emergency radiology.

The following cut-off values are used for AA diagnosis: diameter of appendix ≥7 mm, wall thickness of the appendix ≥2 mm, with possible secondary signs of free fluid in right iliac fossa, lymphadenopathy, and fat stranding.

Different scanning protocols were used: one-phase non-enhanced CT (18.4%, *n* = 85); one-phase late portal CT (55.7%, *n* = 258); two-phase non-enhanced and early arterial CT (0.4% *n* = 2); two-phase non-enhanced and late portal CT (13.4%, *n* = 62); three-phase non-enhanced, late arterial and late portal CT (18.4%, *n* = 85); four-phase non-enhanced, late arterial, late portal and delayed CT (0.6%, *n* = 3). Use of CT was at the discretion of the radiologist.

### 2.2. Surgery and Histology

Negative appendectomy in this study was defined as a surgically removed histologically normal appendix or a diagnostic laparoscopy where the appendix had no visual inflammatory changes and was not removed.

Surgical and histological criteria of catarrhal, phlegmonous and gangrenous appendicitis are listed in Table 1 [9].

### 2.3. Statistical Analysis

The collected data were anonymized and statistically analyzed with IBM SPSS ver. 22.0 Armonk, NY: IBM Corp. Qualitative variables are reported in absolute frequency and percentage, and quantitative variables are reported as medians and interquartile ranges (IQR). We calculated the sensitivity and specificity of the score in the studied population, as well as its positive and negative predictive values. The pre-established confidence interval was 95%, *p* = 0.05. Receiver operating characteristic (ROC) curves correspond to binary logistic regression when the dependent variable is pathological appendiceal changes (0, no appendiceal changes; 1, pathological appendiceal changes).

## 3. Results

A total of 1855 patients were included in the study (Figure 1): 1206 (65.0%) were women, 649 (35.0%) were men, with a median age of 34 years (IQR, 24.5–51). Patient characteristics are presented in Table 2.

TUS was performed in almost all patients (*n* = 1851, 99.8%), while CT was performed in 463 (25.0%) patients. In most cases, the appendix was not visualized with TUS (*n* = 1320, 71.3%); a normal appendix was seen in 144 (7.8%) cases and AA in 279 (15.0%). A total of 108 (5.8%) were classified as probable AA, while 43 (39.8%) of these patients with probable AA underwent a follow-up CT examination (Table 3).

In total, 129 (27.8%) Patients out of 464 with later confirmed AA that were not diagnosed with TUS, both early and advanced disease was missed (Table 4). Suspicion of AA was described in a wide range of AA stages, though in most cases (*n* = 49, 76.6%) it was a less advanced disease. Overall, 64 (59.3%) out of 108 suspicious findings on TUS were proven to be AA. In three cases, TUS showed false-negative results.

In total, out of the 172 (37.1%) of 463 patients diagnosed with AA after CT scan, 42 (9.1%) revealed alternative emergency diagnosis and 249 (53.8%) revealed scans without acute abnormalities. Overall, 249 (53.8%) CT scans showed no urgent pathology. There were nine false-negative CT examinations.

Overall, 464 appendectomies were performed, 4 patients required percutaneous drainage and 24 remained under observation and no interventions were needed. Most appendectomies were laparoscopic (*n* = 437, 94.2%), and only a few open appendectomies were performed (*n* = 27, 5.8%). 

In total, 27 (4.2%) unnecessary operations were performed, revealing normal appendix. An amount of 13 (0.7%) patients were reported to have AA on TUS, and 13 (2.8%) patients were reported to have AA on the CT scan. One patient was operated on based on clinical findings alone. All of the 27 patients underwent diagnostic laparoscopy only. They completely recovered with no repeated interventions within 6 months. There were no cases of removed histologically proven normal appendix.

Four (0.9%) cases of appendiceal carcinoma were confirmed on pathological examination. They were missed by both radiologists and surgeons. They were mostly radiologically reported as AA in three cases and as normal appendix in one case. Again, it is important to note that there were no cases of histologically proven normal appendix removed.

The overall diagnostic imaging performance described by the sensitivity, specificity, and positive and negative predictive values of TUS and CT are shown in Table 5; it was decided to include possible AA into the AA group while counting these values. 

The results of the statistical analysis of the effectiveness of different diagnostic tests, including laboratory test results, are presented in Figure 2. The diagnostic efficacy of the CT scan to correctly diagnose AA is significantly higher than the US imaging or laboratory markers. 

## 4. Discussion

We found that the conditional CT strategy results in a low number of unnecessary surgeries of 4.2%; CT was avoided in 75% of patients. The appendix is visible on TUS in 27.8% of cases: AA is diagnosed in 15%, normal appendix in 7.8% and 5.8% remain equivocal, where follow-up CT is efficient. CT has the highest diagnostic accuracy in patients with suspected acute appendicitis, even though up to 53.8% (*n* = 249) of CT scans did not reveal any other urgent pathological findings.

This study benefits from a large quantity of patient data, collected on a prospective database over a short period of time, with a follow-up 6 months after the initial visit. All patients with clinically suspected AA were included in the study, thus it represents the “real world” data of AA diagnosis in tertiary settings.

The main drawback of the study is the single tertiary center setting, so the applicability of the results may be limited in other environments. Ultrasound investigations may be of different quality due to the examination technique, operator skill, and availability of skilled radiologists or ultrasound technicians during the night-shift hours [10,11].

The main indicator of successful diagnostic workflow of AA is a low negative appendectomy (NA) rate. NA is associated with excess mortality—mortality that is almost at the same level as among patients with perforated appendicitis [12]. Furthermore, NA is significantly associated with an increased risk of ectopic pregnancy [13]. The previously accepted NA rate was around 20% [14]. In our institution, our previous experience of diagnosing AA with clinical examination and TUS resulted in an NA rate of 22.9% [8]. Moreover, the accepted NA rate was even higher in pregnant women [9]. However, recent reports suggest that this number can be significantly and safely reduced with the use of magnetic resonance imaging (MRI) [15]. An increase in CT imaging shows an inversely proportional decrease in NA rate [16]. CT seems to be commonly used both in adults (83%) and in children (73%) with AA in hospitals in the United States [17], despite the increased risk of hematologic malignancies in young adults related to ionizing radiation [18]. Recent studies by Yeh et al. and by Sugiura K et al. included patients with already diagnosed acute appendicitis, where their reported NA rate was 10% and 2.5% with 85% and 95% use of preoperative CT. Their experience of US use was very limited—14% and 19%, respectively [19,20]. Our results, which highlight the importance of the conditional CT strategy, are similar to a previous study [21] where the TUS was able to identify the appendix in 53% of patients. The conditional CT strategy there resulted in a similar sensitivity of 96% vs. 95% when compared with the direct CT group, but a lower specificity of 77% vs. 87%. A recent prospective study employed observation and the conditional CT strategy in patients with suspected AA [22]. They found that the sensitivity and specificity of TUS were 58.2% and 97.3%, respectively, and their model of conditional CT resulted in a low negative surgery rate of 5.8% and a CT scan rate of 19.7%. Routine CT use may be more important in older adult populations, where the risk of future malignancies is less significant [23]. In these situations, routine CT increases the accuracy of the diagnosis and appropriate management in elderly patients.

TUS is a good diagnostic tool to confirm AA—in a study of 3607 patients who underwent appendectomy, TUS was indeterminate in only 30% [11]. The majority (63.6%) of AA cases in the present study were diagnosed with TUS, and there was a low rate of false-positive cases (*n* = 52, 2.8%). Unfortunately, TUS was not effective in visualizing a normal appendix (7.8%), making it a poor tool in ruling out AA. Our results are similar to previous studies, which have reported the capability of TUS to visualize healthy appendix either in adults or in children in 4–6% of cases [24,25]. A suggestion was made that patients with non-visualized appendix on TUS, but otherwise normal scans, are at a significantly lower risk of appendicitis, and active clinical observation should be considered in these patients, rather than a direct referral for CT [26]. However, in our experience, TUS failed to visualize the appendix in 27.8% of AA cases of both early and advanced disease, and we suggest that in cases when the appendix is not visible on TUS, the interpretation should not be made that there is no AA. 

Suggestions were made that repeated TUS may show better diagnostic performance compared to the initial TUS as the progression of the inflammatory process in the appendix would make it easier to detect [4]. The role of observation and repeated laboratory and clinical examination is beyond the scope of this study, and should be studied in future, as currently used prognostic scores and laboratory tests can be misleading and have lower sensitivity and specificity than diagnostic imaging, especially in early disease [27,28,29].

It is interesting to note that 5.8% (*n* = 108) of TUS examinations in our study were inconclusive, suggesting probable AA diagnosis, while 39.8% of patients of this group were followed up with conditional CT; in this latter group the NA rate was four times lower than the TUS-only group (10.8% vs. 2.3%). A recent study revealed that negative or indeterminate TUS might be closely related to female gender, with age > 30 years and higher BMI [11]. Leeuwenburgh et al. developed a “clinical decision rule“ based on gender, clinical symptoms and laboratory results, helping reduce the probability of appendicitis without further imaging after inconclusive ultrasound, from 20% to 6% with an NPV of 94% (95% confidence interval (CI) = 87% to 98%) [30]. The other more concerning issue is the TUS conclusion of “probable appendicitis” (5.8%) that was pathologically proven as AA in about 2/3 (59.3%) of patients. There were no inconclusive cases in the CT group. Two prospective trials of CT vs. TUS in probable appendicitis had similar findings: indeterminate imaging results were significantly more frequent with TUS than with CT [31,32].

Even though clinical scoring systems (i.e., the Alvarado score and AIR) are not accurate enough to be used alone for diagnosis, they might offer the potential benefit of risk stratification and standardization of the initial diagnostic workup [33]. Anderson et al. in their prospective randomized trail summarized that AIR score-based risk classification can safely reduce the use of diagnostic imaging [34]. This could be especially useful for patients after an initial inconclusive ultrasound.

It is interesting to note that CT scans had higher false-positive rates than the TUS (2.8% vs. 0.7%). Atema et al. found that more straightforward cases of appendicitis can adequately be detected by ultrasound comparable to the CT scan [21].

Our study resulted in low NA rates by using a conditional CT protocol and performing CT scans in only 25.0% of patients. However, it is important to note that up to 53.8% (*n* = 249) of CT scans did not reveal any other urgent pathological findings, and patients received unnecessary radiation exposure. Consequently, better selection criteria for CT are needed, as this would further lower patient exposure to radiation and also lower overall hospital costs. Risk stratification with clinical scores could be a potential solution and a possible topic for research in the future. Other options for lowering patient exposure to radiation and the nephrotoxic effect of the intravenous contrast are to change CT protocols to non-contrast CT scans, one-portal venous phase CT scans, or low-dose CT scans [35]. The main drawback of the non-contrast CT imaging is that most of the patients referred to CT for suspected AA are of a young age and low BMI, meaning low amounts of intraabdominal fat, which makes the appendix difficult to find in most cases. Additional contrast-enhanced scans have to be performed, leading to increased radiation doses for patients. Low-dose CT scans in recent meta-analysis show sensitivity and specificity results equal to standard CT protocol [36,37].

An alternative strategy to reduce the rate of unnecessary CT scans after inconclusive or negative TUS results is observation, which is recommended by the updated World Society of Emergency Surgery Jerusalem guidelines [6] for low-risk patients, and was proven to be effective in child populations in previous studies [38]. It is known that inflammatory changes progress over time and a repeated TUS could more easily and confidently identify these changes. Moreover, clinical and laboratory dynamics over time could hypothetically increase the accuracy of diagnosis without using additional sources of ionizing radiation. This topic is the scope of our further ongoing research in a randomized clinical trial being carried out in our institution. 

Unfortunately, this study did not analyze the possibility of differentiating the diagnostic features of complicated vs. non-complicated AA, as recent systematic reviews and meta-analyses of RCTs have concluded that the majority of patients with non-complicated AA can be treated with an antibiotic-first approach, but this requires careful patient selection (6). A complex analysis of laboratory and diagnostic imaging features might enable a more precise and successful selection for treatment strategies, and this could be the scope for further research. Finally, a detailed analysis of acute appendicitis imaging features, especially in the appendix of normal diameter, would of course be a useful topic for future research.

## 5. Conclusions

Imaging is crucially important for the successful diagnosis of AA. The conditional CT strategy is significantly more effective in reducing the negative appendectomy rate in comparison with TUS imaging, resulting in a low NA rate and decrease in CT scan use. Further research for optimal selection criteria for the CT scan in suspected AA should be performed. 

## Figures and Tables

**Figure 1 jcm-10-02456-f001:**
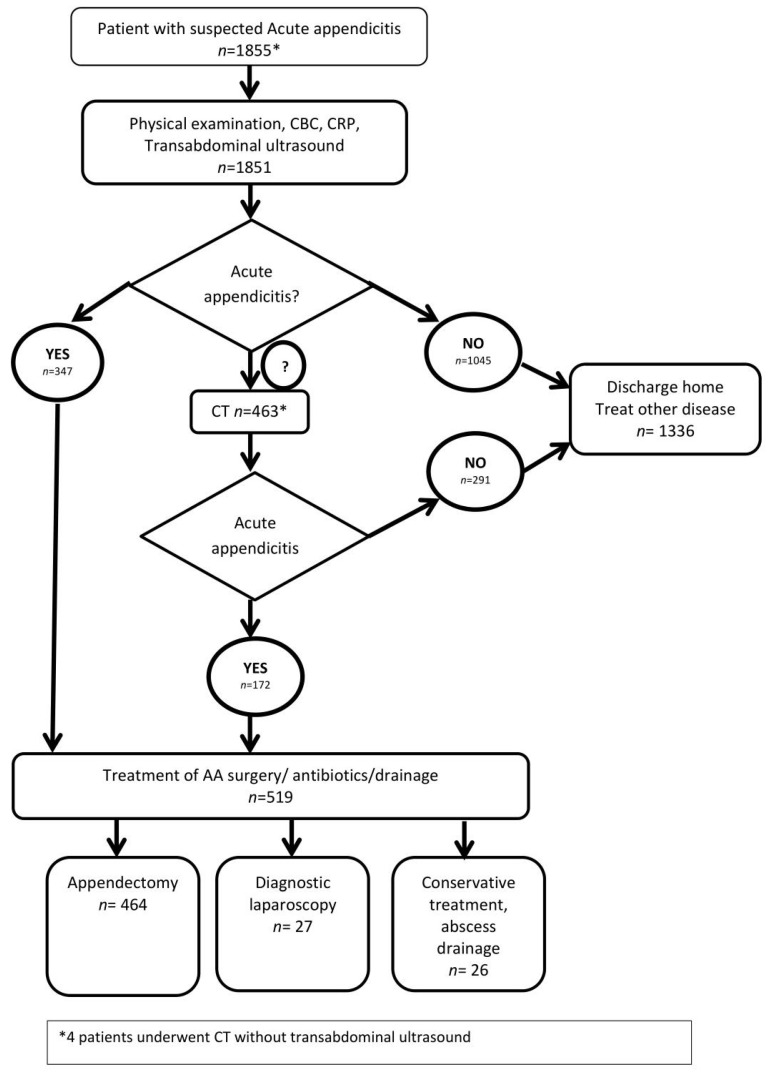
Flowchart of conditional CT-based strategy. CBC—common blood count; CRP—C reactive protein; AA—acute appendicitis.

**Figure 2 jcm-10-02456-f002:**
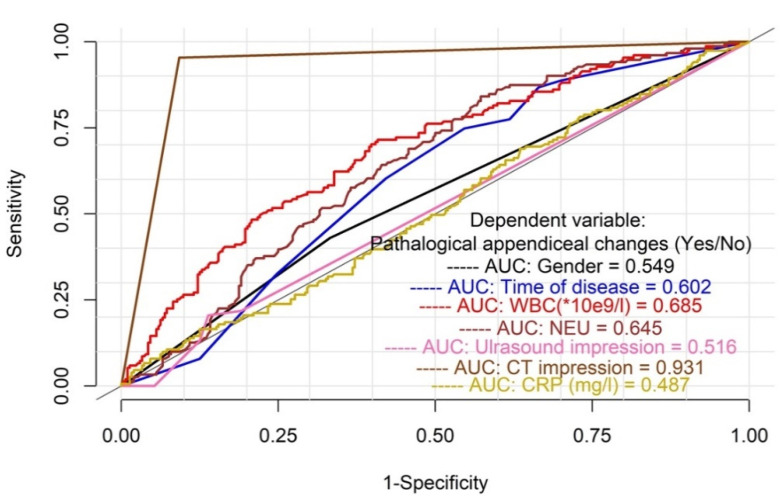
Analysis of effectiveness of different diagnostic tests including laboratory test results. AUC—area under the curve.

**Table 1 jcm-10-02456-t001:** Surgical and histological criteria of appendix changes.

Characteristics	Surgical (Gross) Criteria	Histological (Microscopic) Criteria
Catarrhal appendicitis	No visible changes	Neutrophils within mucosa and mucosal ulceration, with or without intraluminal neutrophils
Secondary changes/periapendicitis	May appear normal or serosa may be dull, congested and show exudate	Inflammation of serosa and subserosa, infiltration extends no further than outer muscularis propria
Phlegmonous appendicitis	Dilated or increased diameter appendix; dull serosa; dilatation and congestion of surface vessels; fibrinopurulent serosal exudate	Neutrophilic infiltration of mucosa, submucosa and muscularis propria; transmural inflammation; extensive ulceration and intramural abscesses; vascular thrombosis
Gangrenous appendicitis	Appendiceal wall friable; purple, green or black	Transmural inflammation with areas of necrosis, extensive mucosal ulceration

**Table 2 jcm-10-02456-t002:** Patient characteristics. TUS—transabdominal ultrasound, CT—computed tomopraghy, AA—acute appendicitis.

Patient Characteristics		*n*	Percentage
**Overall**		1855	100%
Age	18–25	512	27.6%
	26–35	466	25.1%
	36–45	298	16.1%
	46–55	205	11.1%
	56–65	137	7.4%
	66–75	114	6.1%
	76–85	95	5.1%
	>85	28	1.5%
Sex	Women	1206	65%
	Men	649	35%
Transabdominal Ultrasound	Overall	1851	99.8%
	Acute appendicitis	231	12.5%
	Perforated acute appendicitis	4	0.2%
	Acute appendicitis with periappendiceal abscess	28	1.5%
	Acute other disease	93	5.0%
	Suspected acute appendicitis	108	5.8%
	Normal appendix	144	7.8%
	TUS not done	4	0.2%
Computed Tomography	Overall	463	25.0%
	Acute appendicitis	145	7.8%
	Perforated acute appendicitis	8	0.4%
	Acute appendicitis with periappendiceal abscess	19	1.0%
	Other disease	42	2.3%
	Normal appendix	291	15.8%
	CT not done	1392	75.0%
Diagnosis	Uncomplicated acute appendicitis	460	24.9%
	Complicated acute appendicitis	30	1.6%
	Appendiceal carcinoma	4	0.2%
	Other diseases	1332	71.9%
	No identified cause	25	1.4%
Treatment Interventions	Laparoscopic appendectomy	437	23.6%
	Open appendectomy	27	1.5%
	Abscess drainage	4	0.2%
	Conservative management of AA	22	1.2%
	Diagnostic laparoscopy	27	1.5%
Surgical Findings (*n* = 464)	Catarrhal appendicitis	9	1.9%
	Secondary appendicitis	3	0.6%
	Phlegmonous appendicitis	268	57.8%
	Gangrenous appendicitis	184	39.7%
	Cancer	0	
Histopathological Findings (*n* = 464)	Catarrhal appendicitis	7	1.5%
	Secondary appendicitis	6	1.2%
	Phlegmonous appendicitis	301	64.9%
	Gangrenous appendicitis	146	31.5%
	Cancer	4	0.9%

**Table 3 jcm-10-02456-t003:** Outcomes of patients with probable AA after TUS—transabdominal ultrasound.

Characteristics	TUS Only (*n* = 65)	TUS + CT (*n* = 43)	OR	CI (95%)	*p*-Value
Acute Appendicitis	42 (64.6%)	22 (51.2%)	0.5758		
				0.2638–1.2566	0.1653

Negative Surgery	7 (10.8%)	1 (2.3%)	0.2953	0.0683–1.2772	0.1358
Other Diagnosis Confirmed	8 (12.3%)	13 (30.2%)	3.1073	1.1792–8.1866	0.0249
Observation	8 (12.3%)	7 (16.3%)	1.3896	0.4584–4.2118	0.5602

**Table 4 jcm-10-02456-t004:** TUS and CT in patients with later confirmed AA.

		Pathological Diagnosis (*n* = 464)					OVERALL
		Catarrhal Appendicitis	Secondary Changes	Phlegmonous Appendicitis	Gangrenous Appendicitis	Cancer	
TUS Findings (*n* = 462)	Normal appendix	1 (0.2%)	0 (0%)	2 (0.4%)	0 (0%)	0 (0%)	3 (0.6%)
	AA	1 (0.2%)	3 (0.7%)	180 (38.8%)	82 (17.7%)	2 (0.4%)	266 (57.6%)
	Possible AA	3 (0,7%)	1 (0.2%)	46 (10.0%)	14 (3.0%)	0	64 (13.9%)
	Not visualized	2 (0.4%)	2 (0.4%)	73 (15.7%)	50 (11.0%)	2 (0.4%)	129 (27.9%)
CT Findings (*n* = 160)	Normal appendix	0 (0%)	1 (0.6%)	5 (3.1%)	2 (1.3%)	1 (0.6%)	9 (5.6%)
	AA	3 (1.9%)	3 (1.9%)	87 (54.4%)	57 (35.6%)	1 (0.6%)	151 (94.3%)

**Table 5 jcm-10-02456-t005:** Sensitivity and specificity for AA of AU and CT.

	TUS	CT
Sensitivity	71.4%	93.8%
Specificity	96.2%	93.9%
Positive Predictive Value	86.6%	88.4%
Negative Predictive Value	90.8%	96.5%

## Data Availability

The datasets used and/or analyzed during the current study are available from the corresponding author on reasonable request.

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
