# Peer review of "Conditional CT Strategy—An Effective Tool to Reduce Negative Appendectomy Rate and the Overuse of the CT"

_jcm, 2021, doi:10.3390/jcm10112456_

Round 1
Reviewer 1 Report
Thank you very much for the opportunity to review the manuscript. The authors submitted a manuscript investigating the effectiveness of conditional CT strategy to reduce negative appendectomy. This retrospective study two main weaknesses that have to be considered.
First, negative appendectomy rate of this study (4.2%) is not superior to, and even lower than those reported by recent similar studies (Yeh DD, et al. Multicenter study of the treatment of appendicitis in America: acute, perforated, and gangrenous (MUSTANG), an EAST multicenter study. Ann Surg. 2021;273(3):548-556. Sugiura K, et al. Chronological changes in appendiceal pathology among patients who underwent appendectomy for suspected acute appendicitis. World J Surg. 2020;44(9):2965-2973.) Their conditional CT strategy cannot be validated based on the rate.
Second, the surgical and histological criteria of catarrhal, secondary, phlegmonous, and gangrenous appendicitis.
In addition, the number of perforated appendicitis that is most clinically important was nor described.
Third, patients flow is so confused that readers cannot easily understand the analysis.
Following points should be addressed:
p.2. The wall structure and contour of the appendix was evaluated? These findings is important for diagnosis of gangrenous and perforated appendicitis.
p.3. CT scanning protocol was described, however, how was the rate of contrast enhancement of CT?
p.3. Figure 1. The flow chart should include the number of appendectomy, laparoscopy only, and follow up.
p.4. Table 1. The diagnosis of TUS and CT should include perforated appendicitis and periappendiceal abscess.
p.4. Table 1. Present reviewer would suggest to describe the surgical and histological criteria of catarrhal, secondary, phlegmonous, and gangrenous appendicitis.
p.5. CT was performed in 25.0% of suspected appendicitis. The low rate can be a cause of relatively high rate of negative appendectomy (4.2%). The studies of Yeh DD, et al. and Sugiura K, et al reported over 90% of CT performed, and below 3% of negative appendectomy rate.
p.5. Table 2. What is the purpose of the Table?
p.5. The appendix was not visualized in 71.3% of cases. Why the rate was so high?
p.6. (line 143) 27(4.2%) unnecessary operations were performed revealing normal appendix. (line 150) It is important to note 150 that there were no cases of histologically proven normal appendix was removed. These two sentences is contradictory.
p.6. Figure 2. The number of ROC curves was so many to understand. Arrange them to more important factors. Present reviewer would suggest to describe the method to create the ROC curves.
Reviewer 2 Report
The authors describe their strategy of using a conditional CT strategy in 1855 patients with possible appendicitis at a single institution.
They conclude that the strategy and use of ultrasound was effective in reducing negative appendectomy rate and avoids unnecessary CT scans in a large proportion of patients.
It would be helpful if possible to note the number of patients admitted for observation and serial exams if possible as well (while reducing negative appendectomy rate is helpful, if substituted for longer length of stays may have some tradeoffs that are negative).
This could also be noted in the discussion.
Author Response
Response to Reviewer 2 Comments
Thank you so much for your comments and suggestions for improvements.
Point 1: It would be helpful if possible, to note the number of patients admitted for observation and serial exams if possible as well (while reducing negative appendectomy rate is helpful, if substituted for longer length of stays may have some tradeoffs that are negative).
Response 1: Unfortunately, at the period of the study we did not practice observation, so this data was not collected. We had 22 cases that were treated conservatively using antibiotic therapy. These patients did not have a recurrence over 30-day follow-up. However, we are now concluding an ongoing prospective randomized study that analyses the effectiveness of observation in diagnostics of medium and low appendicitis risk group patients. We expect to have the results in few months.
Point 2: This (role of observation) could also be noted in the discussion.
Response 2: We expanded on the role of observation in our discussion (lines 219-222; 232-235; 282-290).